# BRAF status modulates Interelukin-8 expression through a CHOP-dependent mechanism in colorectal cancer

Fabiana Conciatori[1,8], Chiara Bazzichetto [1,8], Carla Azzurra Amoreo[2], Isabella Sperduti[3], Sara Donzelli[4], Maria Grazia Diodoro [2], Simonetta Buglioni[2], Italia Falcone [1], Senji Shirasawa[5], Giovanni Blandino [4], Gianluigi Ferretti [1], Francesco Cognetti[1], Michele Milella[6,8 ✉] & Ludovica Ciuffreda [1,7,8]

Inflammation might substantially contribute to the limited therapeutic success of current systemic therapies in colorectal cancer (CRC). Amongst cytokines involved in CRC biology, the proinflammatory chemokine IL-8 has recently emerged as a potential prognostic/predictive biomarker. Here, we show that *BRAF* mutations and PTEN-loss are associated with high IL-8 levels in CRC models in vitro and that BRAF/MEK/ERK, but not PI3K/mTOR, targeting controls its production in different genetic contexts. In particular, we identified a BRAF/ERK2/CHOP axis affecting IL-8 transcription, through regulation of CHOP subcellular localization, and response to targeted inhibitors. Moreover, RNA Pol II and an open chromatin status in the CHOP-binding region of the IL-8 gene promoter cooperate towards increased IL-8 expression, after a selective BRAF inhibition. Overall, our data show that IL-8 production is finely and differentially regulated depending on the tumor genetic context and might be targeted for therapeutic purposes in molecularly defined subgroups of CRC patients.

[1] Medical Oncology 1, IRCCS - Regina Elena National Cancer Institute, Via Elio Chianesi 53, 00144 Rome, Italy. [2] Department of Pathology, IRCCS - Regina Elena National Cancer Institute, Via Elio Chianesi 53, 00144 Rome, Italy. [3] Department of Biostatistics, IRCCS - Regina Elena National Cancer Institute, Via Elio Chianesi 53, 00144 Rome, Italy. [4] Oncogenomic and Epigenetic Unit, Molecular Chemoprevention Group, Department of Research, Diagnosis and Innovative Technologies, IRCCS - Regina Elena National Cancer Institute, Via Elio Chianesi 53, 00144 Rome, Italy. [5] Central Research Institute for Advanced Molecular Medicine, Fukuoka University, 8 Chome-19-1 Nanakuma, Jonan Ward, Fukuoka, Japan. [6] Section of Oncology, Department of Medicine, University of Verona School of Medicine and Verona University Hospital Trust, Piazzale Aristide Stefani 1, 37126 Verona, Italy. [7] SAFU, Department of Research, Advanced Diagnostics, and Technological Innovation, IRCCS Regina Elena National Cancer Institute, Rome 00144, Italy. [8] These authors contributed equally: Fabiana Conciatori, Chiara Bazzichetto, Michele Milella, Ludovica Ciuffreda. ✉email: michele.milella@univr.it

Cytokine networks contribute to the development and progression of cancer, particularly in Colorectal Cancer (CRC), in which inflammation represents a critical aspect of disease progression[1]. Within the Tumor MicroEnvironment (TME), both stromal and cancer cells release cytokines/chemokines and growth factors, thereby contributing to the cytokine networks, which modulate the inflammatory/immunologic milieu of cancer tissues[2].

InterLeukin (IL)-8, also referred to as CXCL8, is a proinflammatory CXC ELR + chemokine; through the binding to its cell-surface G protein-coupled receptors, CXCR-1 and CXCR-2, IL-8 plays multiple roles in cancer, driving the activation of key signaling pathways in both stromal and intestinal epithelial cells, to promote or increase proliferation, angiogenesis and metastasis[3]. Recent evidence has shown a correlation between IL-8 overexpression and both Vascular Endothelial Growth Factor (VEGF)-independent tumor angiogenesis and chemoresistance in preclinical models, thus highlighting a role for IL-8 expression in CRC[4]. High levels of IL-8 are observed in the serum and cancer tissue of CRC patients and these levels significantly increase according to a worsening clinical stage and tumor grade[5–7]. A recent meta-analysis has suggested that high levels of IL-8 expression are significantly associated with poor prognosis in CRC patients (HR = 1.54, 95% CI 1.03–2.32) and correlate with advanced stage, lymphatic and liver metastasis, and resistance to antiangiogenic agents[8,9].

IL-8 expression can be regulated at the transcriptional and post-transcriptional levels by multiple intracellular signaling pathways, including the Mitogen-Activated Protein Kinase (MAPK) and PhosphoInositide 3-Kinase (PI3K)[10,11]. However, the molecular mechanisms and transcription factor networks through which signaling pathways regulate IL-8 expression in specific, genetically defined, cancer contexts remain to be defined.

In this study, we have investigated the relationships between BRAF mutations/loss of Phosphatase and TENsin homologue deleted on chromosome 10 (PTEN) and IL-8 production and found that both alterations contribute to high levels of IL-8 production in a panel of genetically characterized CRC cell lines. IL-8 production was profoundly influenced by modulation of the MAPK pathway: in particular, BRAF inhibition by dabrafenib abrogated IL-8 production in BRAF-mut cell lines, but paradoxically increased it in BRAF-wt contexts, while MEK inhibition and Extracellular signal-Regulated Kinase (ERK)2 silencing selectively abrogated IL-8 production, regardless of the genetic background of the CRC cell lines examined. Double PI3K/ mammalian Target Of Rapamycin (mTOR) inhibition, on the other hand, did not substantially affect IL-8 production. Dabrafenib-induced IL-8 modulation was found to be mechanistically related to nuclear export of the C/EBP HOmologous Protein (CHOP) transcription factor in BRAF-mut cells and to CHOP nuclear retention and promoter binding in BRAF-wt contexts. Overall, our data highlight a BRAF/ERK2/CHOP regulatory axis which regulates both basal and drug-induced IL-8 expression in CRC models.

## Results

### BRAF mutations and PTEN-loss correlate with IL-8 production. 
We investigated the role of the MAPK and PI3K pathways in the regulation of IL-8 expression in a panel of 28 CRC cell lines, characterized for BRAF, Kirsten RAt Sarcoma (KRAS), PTEN and PI3K gene status (Table 1 and Supplementary Table 1). PTEN expression was also investigated at the mRNA and protein level (Supplementary Fig. 1 and Table 1): PTEN protein expression was completely absent in 12 CRC cell lines, moderate in 7, and strong in 9 of the tested cell lines. Cell lines carrying PTEN deletions or inactivating mutations or completely lacking PTEN protein expression are referred to as PTEN-loss[12].

Cell culture media of the 28 CRC cell lines (BRAF-mut = 12; KRAS-mut = 9; PI3K-mut = 13; PTEN-loss = 12, Table 1 and Supplementary Table 1) were analyzed by ELISA assay under standardized culture conditions (see Methods) (Fig. 1a). Statistical analysis showed a statistically significant correlation between IL-8 expression and BRAF status: indeed, the presence of a BRAF[V600E] mutation predicted IL-8 levels higher than 257.5 pg/mL with 58.33% sensitivity and 93.75% specificity (Area Under the Curve (AUC) = 0.76) and the Receiver Operating Characteristic (ROC) curve-based prediction algorithm based on BRAF mutation had 52% accuracy in predicting IL-8 production ($p = 0.004$) (Fig. 1b). Statistical analysis also showed a trend towards a statistically significant association with PTEN-loss: indeed, PTEN-loss predicted an IL-8 value higher than 42.5 pg/mL with a 64.29% sensitivity and 78.57% specificity (AUC = 0.71) and the ROC curve-based prediction algorithm based on PTEN-loss had 43% accuracy in predicting IL-8 production ($p = 0.05$) (Fig. 1b). Moreover, the highest levels of IL-8 were observed in CRC cell lines carrying both BRAF[V600E] and PTEN-loss ($n = 5$): indeed, combined BRAF/PTEN analysis predicted an IL-8 value higher than 46 pg/mL with 87.50% sensitivity and 80.00% specificity (AUC = 0.88) and the ROC curve-based prediction algorithm based on these two alterations had 68% accuracy in predicting IL-8 production ($p = 0.002$) (Fig. 1b). Conversely, KRAS and PI3K mutation status were not correlated with IL-8 expression (Supplementary Fig. 2).

We also confirmed the specific correlation between IL-8 expression and PTEN status by using X-MAN™ isogenic HCT116 cell lines (HCT116 and HCT116 PTEN$^{-/-}$). To this purpose, cell culture media of X-MAN™ isogenic HCT116 cell lines were analyzed after 24 h of culture in serum-free medium by Human Angiogenesis Antibody Array, which revealed the selective expression of IL-8 only by HCT116 PTEN$^{-/-}$ (Fig. 1c). This result was confirmed by IL-8 specific ELISA assay (Fig. 1d).

In order to verify the specific correlation between BRAF/PTEN status and IL-8 production, we also evaluated the levels of VEGF and IL-6, two other pro-angiogenic soluble factors involved in CRC progression. Differently from IL-8 production, no significant correlation was observed between BRAF status and VEGF production (Supplementary Fig. 3). However, in normoxic conditions of growth, PTEN-loss predicted VEGF levels higher than 621 pg/mL with a 92.86% sensitivity and 57.14% specificity (AUC = 0.77) and the ROC curve-based prediction algorithm based on PTEN-loss had 50% accuracy in predicting VEGF production ($p = 0.01$). KRAS-wt status also predicted VEGF levels higher than 627 pg/mL with a 80% sensitivity and 75% specificity (AUC = 0.75) and the ROC curve-based prediction algorithm based on KRAS-wt had 55% accuracy in predicting VEGF production ($p = 0.01$). IL-6 was not detectable in the culture media of the CRC cell lines examined (Supplementary Fig. 4).

Taken together, these results demonstrate that both BRAF and PTEN status specifically determine IL-8 expression.

### MAPK-dependent regulation of IL-8 expression. 
We next investigated whether specific inhibitors targeting BRAF, MEK, and PI3K/mTOR (dabrafenib, trametinib, and gedatolisib, respectively) could modulate IL-8 expression. To this purpose, four CRC cell lines (SNU1235, SNU1047, HT29, and LS180), differing for BRAF and PTEN status (BRAF[V600E]/PTEN-loss, BRAF-wt/PTEN-loss, BRAF[V600E]/PTEN-competent, BRAF-wt/ PTEN-competent) were exposed to increasing concentration of drugs for 24 h. As shown in Fig. 2a, b, double inhibition of PI3K/ mTOR by gedatolisib minimally affected IL-8 release, regardless

**Table 1 Genetic characterization of 28 CRC cell lines.**

| Cell line | BRAF | KRAS | PTEN | | | PI3K |
|---|---|---|---|---|---|---|
| | | | Gene | mRNA abundance[a] | Protein expression[b] | |
| SNU1544 | wt | wt | wt | 0.21 | + | p.H1047R |
| MDST8 | p.V600K | wt | wt | 0.00 | − | wt |
| OUMS23 | p.V600E | wt | n.a. | 0.01 | − | wt |
| LIM2537 | p.V600E | wt | wt | 0.20 | − | wt |
| LIM2412 | p.V600E | wt | wt | 0.13 | − | wt |
| LS411N | p.V600E | wt | p.C105fs* | 0.10 | − | wt |
| SNUC4 | wt | wt | p.F241S; V290* | 0.62 | − | p.V71I; p.E545G |
| SNUC5 | p.V600E | wt | wt | 0.78 | + | p.H1047R |
| SNU1040 | wt | wt | p.R335*; p.T232A | 0.00 | − | wt |
| SNU1047 | wt | wt | p.K267fs | 0.15 | − | wt |
| SNU1235 | p.V600E | wt | p.R130* | 0.11 | − | wt |
| KM12C | wt | wt | p.G129* | 0.19 | − | wt |
| VACO432 | p.V600E | wt | wt | 0.32 | ++ | p.H1047R |
| HT29 | p.V600E | wt | wt | 0.54 | ++ | wt |
| LS180 | wt | p.G12D | wt | 0.46 | ++ | p.H1047R |
| SW620 | wt | p.G12V | wt | 0.39 | ++ | wt |
| SW480 | wt | p.G12V | wt | 0.12 | + | wt |
| COLO205 | p.V600E | wt | wt | 0.37 | ++ | wt |
| RKO | p.V600E | wt | wt | 0.11 | + | p.I391M; p.H1047R |
| COGA3 | wt | p.G13D | p.R173H | 0.30 | − | wt |
| HROC87 | p.V600E | wt | n.a. | 0.29 | ++ | wt |
| HCT116 | wt | p.G13D/wt | wt | 0.21 | + | p.H1047R |
| HCT116 PTEN⁻/⁻ | wt | p.G13D | del ex5 | 0.11 | − | p.H1047R |
| HKE-3 | wt | wt | wt | 0.36 | + | p.H1047R |
| HK2-6 | wt | p.G12C; p.G13D/G13D | wt | 0.21 | + | p.H1047R |
| DLD-1 | wt | p.G13D/wt | wt | 0.38 | ++ | p.E545K; p.D549N |
| DKO-1 | wt | p.G12C; p.G13D/G13D | wt | 0.33 | ++ | p.E545K; p.D549N |
| DKO-4 | wt | wt | wt | 0.36 | ++ | p.E545K; p.D549N |

[a]Results represent PTEN mRNA abundance relative to positive control T98G
[b]OD ratio of PTEN antibody/β-Actin for each individual sample is compared with OD of positive control T98G. + score 0.01–0.59; + + score 0.6–1. n.a., not available
*Indicates a stop codon

of *BRAF* and PTEN status (see also Supplementary Fig. 5); similar to gedatolisib, selective PI3K (using alpelisib), AKT (using MK226), and mTOR (using everolimus) inhibition decreased IL-8 production by less than 50%, independent of the genetic background of the cell lines tested (Supplementary Fig. 6). Effects of selective BRAF inhibition on IL-8 production were profoundly influenced by *BRAF*-mutational status: dabrafenib strongly inhibited IL-8 production in *BRAF^V600E^* cell lines (SNU1235 and HT29; Fig. 2a), while it increased IL-8 levels in *BRAF*-wt cell lines (SNU1047 and LS180; Fig. 2b); dabrafenib effects on IL-8 production did not differ qualitatively according to PTEN status (Fig. 2a, b and Supplementary Fig. 5). Conversely, MEK inhibition by trametinib (Fig. 2a, b) and ERK1/2 inhibition by SCH772984 (Supplementary Fig. 7) profoundly suppressed IL-8 expression in all the tested cell lines, regardless of their genetic background. As shown in Fig. 2c the combination of dabrafenib and trametinib prevented dabrafenib-induced IL-8 upregulation in *BRAF*-wt contexts (SNU1047 and LS180), but did not further increase IL-8 inhibition as compared to trametinib alone in either genetic context or dabrafenib alone in *BRAF^V600E^* cell lines. Specific MAPK pathway-dependent IL-8 regulation was further confirmed by VEGF expression analysis: indeed, with the notable exception of *BRAF*-wt/PTEN-competent LS180 cell line, VEGF levels under normoxic conditions were much less affected by pathway inhibitors, regardless of the genetic background of the tested cell lines (Supplementary Fig. 8a).

The role of individual MAPK pathway elements was further examined by BRAF, MEK, ERK1, and ERK2 silencing, using short interfering (si) RNA and short hairpin RNA (sh) RNA. As shown in Fig. 3a, IL-8 production was downregulated after BRAF, MEK, or ERK2 silencing, regardless of *BRAF*-mutational status in three of the four analyzed cell lines. More variable effects, namely IL-8 level increase after BRAF silencing and lack of IL-8 decrease after ERK2 silencing, were observed in the LS180 cell line (*BRAF*-wt/PTEN-competent), which also harbors a *KRAS^G12D^* mutation (Table 1 and Supplementary Table 1). ERK1 silencing did not affect IL-8 expression in any of the tested cell lines. VEGF production was consistently less affected by the silencing of MAPK elements, regardless of *BRAF* and PTEN status, thus further suggesting a specific role for the MAPK pathway in the regulation of IL-8 expression in CRC (Supplementary Fig. 8b).

To further analyze the mechanisms of dabrafenib-induced differential IL-8 regulation, we evaluated the effects of dabrafenib in the presence of specific silencing of MAPK elements (Fig. 3b). In *BRAF*-mut cell lines (SNU1235 and HT29), BRAF/ERK1/ERK2 silencing did not substantially modify dabrafenib's inhibitory effects on IL-8 production. Conversely, ERK1 or ERK2 silencing strikingly potentiated dabrafenib-induced IL-8 upregulation in both *BRAF*-wt cell lines (SNU1047 and LS180); BRAF silencing, on the other hand, strikingly potentiated dabrafenib-mediated IL-8 induction in LS180 (which also carry a *KRAS* mutation), but not in SNU1047.

**CHOP-dependent transcriptional regulation of IL-8.** We next analyzed IL-8 mRNA levels after dabrafenib and trametinib treatment by Real Time quantitative Polymerase Chain Reaction (RT-qPCR). As shown in Fig. 4a, mRNA modulation closely paralleled protein expression data: dabrafenib differentially affected IL-8 mRNA expression depending on the genetic context

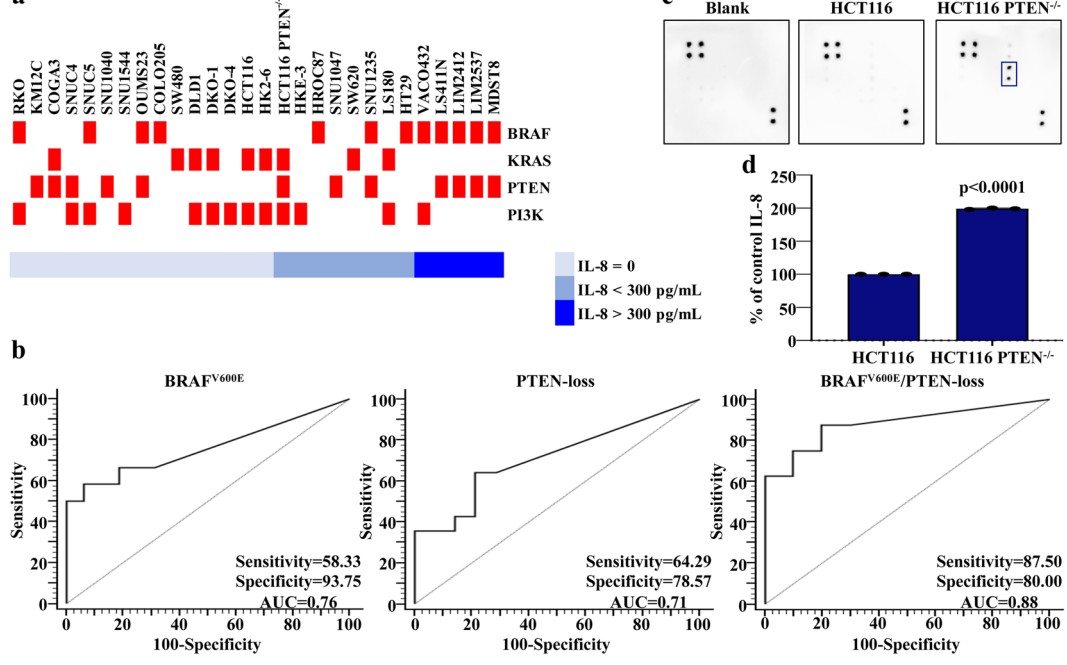

**Fig. 1 IL-8 expression in a panel of 28 CRC cell lines. a** 28 CRC cell lines analyzed for their relative IL-8 expression (shade of blue) and their genetic background of *BRAF, KRAS* and *PI3K* or PTEN protein lack expression (reported in red); the results were expressed as pg/mL for $1 \times 10^6$ cells. **b** Predicting high levels of IL-8 expression according to *BRAF*/PTEN status. **c** Cell culture media of X-MAN™ isogenic HCT116 cell lines (HCT116 and HCT116 PTEN$^{-/-}$) were analyzed by Human Angiogenesis Antibody Array (in the rectangle in blue is highlighted the expression of IL-8). **d** IL-8 expression was confirmed by specific ELISA assay (reported as % of control); results represent the average of three independent experiments. *p*-value was obtained by 2-tailed Student's *t* test for the comparison between parental and PTEN$^{-/-}$ X-MAN™ isogenic HCT116 cell lines.

of the CRC cells (strong down- and upregulation in *BRAF^V600E^* and *BRAF*-wt contexts, respectively), while trametinib downregulated IL-8 mRNA levels regardless of *BRAF* status. Drugs effects on IL-8 transcription were already evident after 2 h of treatment, confirming that dabrafenib and trametinib differentially regulate IL-8 expression by interfering with its transcription. Among transcription factors potentially involved, c-Jun Ser73 phosphorylation and CHOP expression paralleled IL-8 mRNA and protein modulation, in response to BRAF and MEK inhibition (Supplementary Fig. 9a), suggesting a potential involvement of both Activator Protein (AP)-1 and CHOP in IL-8 regulation. To formally prove their involvement, luciferase (luc) genes containing different IL-8 promoter constructs were co-transfected with pRL-TK into the four CRC model cell lines. Consistent with the known role of AP-1 and Nuclear Factor kappa-light-chain-enhancer of activated B cells (NF-κB)/Nuclear Factor for IL-6 expression (NF-IL-6), lack or mutation of NF-κB and NF-IL-6 binding sites reduced luc activity, as compared to a full-length IL-8 promoter (546-luc) (Supplementary Fig. 9b,c)[13]. However, a minimal NF-κB/ NF-IL-6 promoter (98-luc) was not able to sustain basal transcription and the lack of a portion or the complete CHOP-binding sites partially or completely abrogated luc activity, respectively (Fig. 4b, c). Overall, these data confirm the known involvement of AP-1 and strongly indicate CHOP as a relevant player in IL-8 transcription in CRC cell lines, both in basal conditions and in response to MAPK inhibition.

**CHOP localization according to BRAF-selective inhibition.**
According to our hypothesis, in cellular fractionation experiments dabrafenib down- or upregulated CHOP in the nuclear compartment in *BRAF^V600E^* (HT29) or *BRAF*-wt contexts (LS180), respectively (Fig. 5a; see also Supplementary Fig. 10a for

additional data in the SNU1235 and SNU1047 cell lines); conversely, trametinib treatment downregulated CHOP in the nucleus regardless of *BRAF* status (Fig. 5a and Supplementary Fig. 10a). Immunofluorescence experiments confirmed that dabrafenib caused CHOP redistribution to the perinuclear region in the *BRAF^V600E^* context, whereas it upregulated nuclear CHOP in *BRAF*-wt cell lines (Fig. 5b and Supplementary Fig. 10b). MEK inhibition, on the other hand, caused CHOP nuclear exclusion regardless of *BRAF* status (Fig. 5b and Supplementary Fig. 10b).

**CHOP and RNA Pol II mediate IL-8 transcriptional activation.**
We next analyzed the potential role of CHOP-mediated IL-8 transcription in response to pharmacological MAPK inhibition. A constitutive physical interaction between CHOP and RNA Polymerase II (RNA Pol II) was observed regardless of the pharmacological treatment, as assessed by co-immunoprecipitation experiments conducted in the LS180 *BRAF*-wt cell line (Fig. 6a). CHOP and RNA Pol II recruitment to the IL-8 promoter in response to dabrafenib and trametinib treatment was further studied using Chromatin ImmunoPrecipitation (ChIP) assays. As shown in Fig. 6b, recruitment of CHOP to the CHOP-binding site of the IL-8 promoter did not change significantly upon drug treatment; however, selective BRAF inhibition by dabrafenib specifically increased the recruitment of RNA Pol II to the CHOP-binding site of the IL-8 promoter, without increasing its CHOP-independent binding to the IL-8 promoter TATA box (Fig. 6c, d). Moreover, a marked increase in the CHOP-binding site activity of the IL-8 promoter in response to dabrafenib, but not in response to trametinib, was observed after ChIP with an anti-acetylated histone H4 (Fig. 6e), suggesting increased IL-8 promoter accessibility to the CHOP-RNA Pol II complex in response to selective BRAF inhibition in a *BRAF*-wt context.

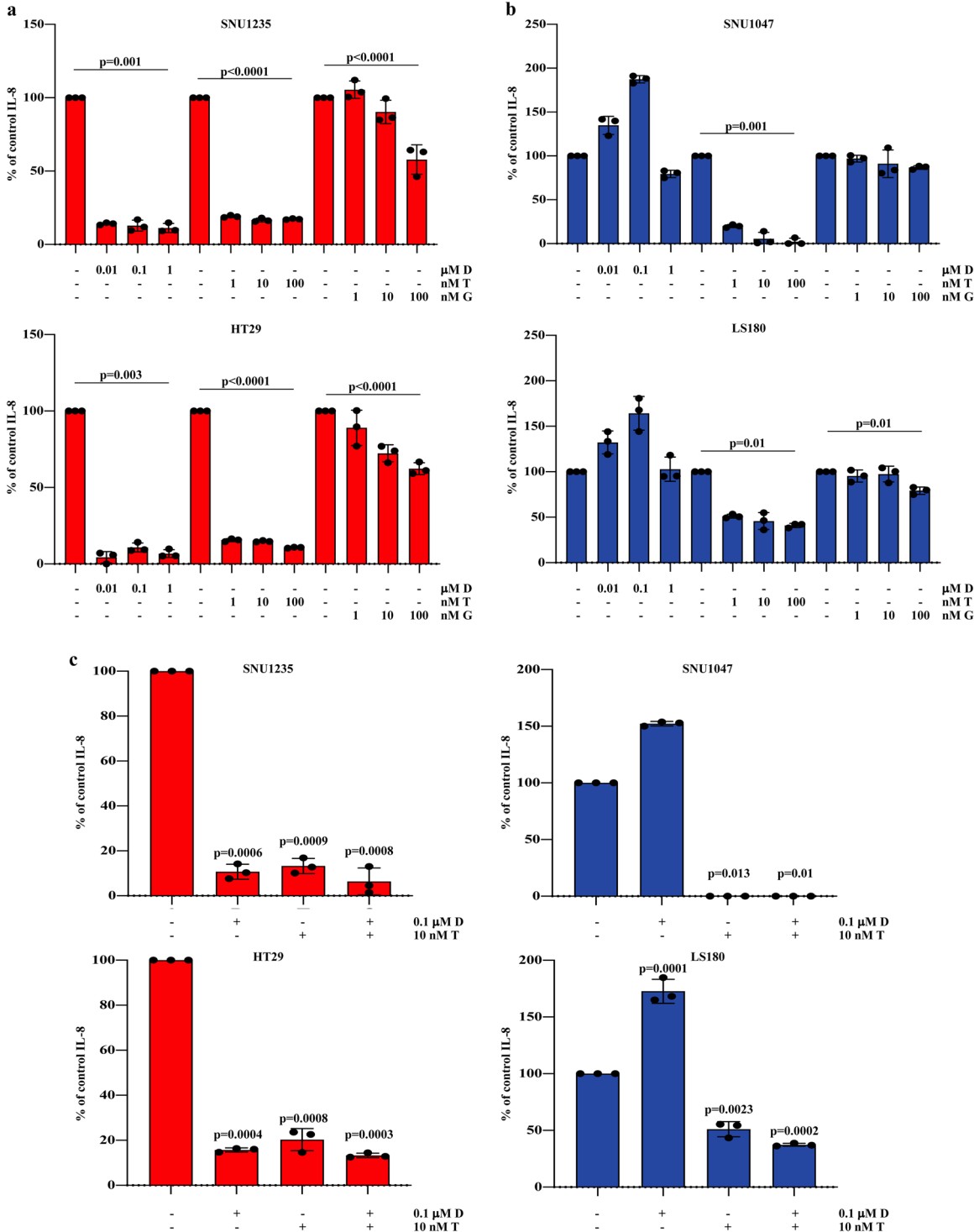

**Fig. 2 Effects of MAPK or PI3K/mTOR inhibition on IL-8 expression.** SNU1235, SNU1047, HT29, and LS180 cell lines were treated with increasing concentration of drugs (dose range 0.01–1 μM dabrafenib (D), 1–100 nM trametinib (T) and 1–100 nM gedatolisib (G)) (**a**, **b**) or with D in combination with T at fixed doses (**c**), as indicated. IL-8 expression was measured after 24 h of treatment, by IL-8 ELISA assay; the percentage of IL-8 was obtained from pg/mL assuming the levels in control cells as 100%. Results represent the average of three independent experiments. For statistical analysis, ANOVA (**a**, **b**) or 2-tailed Student's *t* test for the comparison between control and treated samples (**c**) were used as appropriate.

Using a CHOP-defective IL-8 promoter-luc construct, we further confirmed CHOP-mediated transcriptional activation of the IL-8 gene in response to dabrafenib (Fig. 6f): indeed, dabrafenib (but not trametinib) consistently induced luc transcription in LS180 cells transfected with the complete IL-8 promoter-luc construct (546-luc, see also Fig. 4b), while such response was abrogated in

the presence of a CHOP-defective promoter construct (CHOP mut-luc, see also Fig. 4b).

## Discussion
Our data show that *BRAF* mutations and PTEN-loss promote (in a non-mutually exclusive fashion) high levels of constitutive IL-8

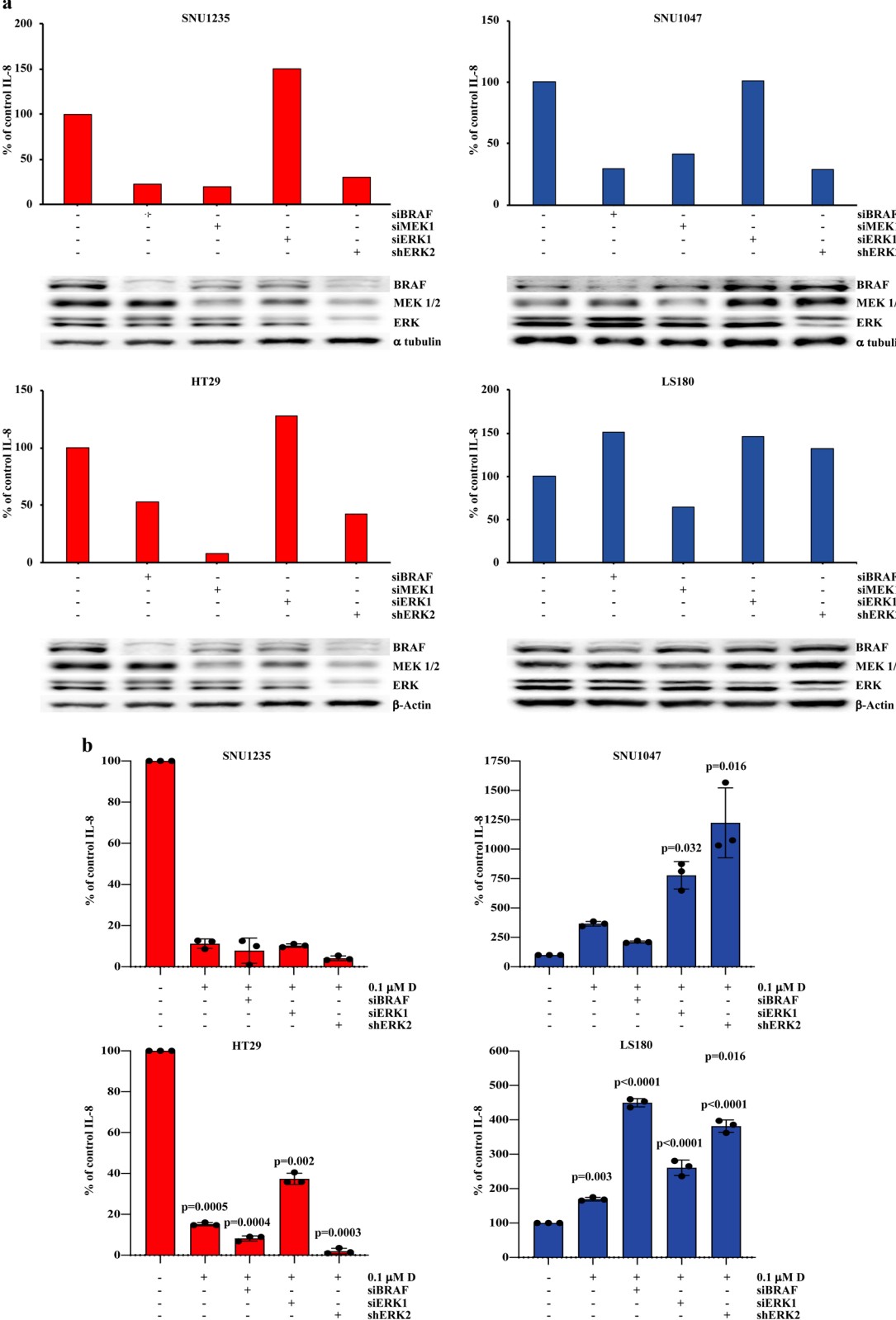

**Fig. 3 Analysis of MAPK elements in affecting IL-8 expression.** In SNU1235, SNU1047, HT29 and LS180 cell lines, BRAF, MEK, ERK1 and ERK2 were knocked down by transient transfection of RNA interference for 24 h, alone (**a**) or in combination with a dose fixed of D (**b**). Molecular effects on protein expression was analyzed by Western Blot (WB) using specific antibodies (β-Actin and α tubulin are shown as protein loading and blotting control). IL-8 was measured after 24 h of culture in serum-free medium, using IL-8 ELISA; IL-8 levels were measured as pg/mL and results are expressed as % of untreated control levels. Results of a representative experiment out of three independent experiments performed (**a**) or the average of three independent experiments (**b**) are shown. p-values indicate statistically significant differences ($p < 0.05$ by 2-tailed Student's $t$ test) for the comparison between treated/silenced and non-treated/silenced conditions of growth.

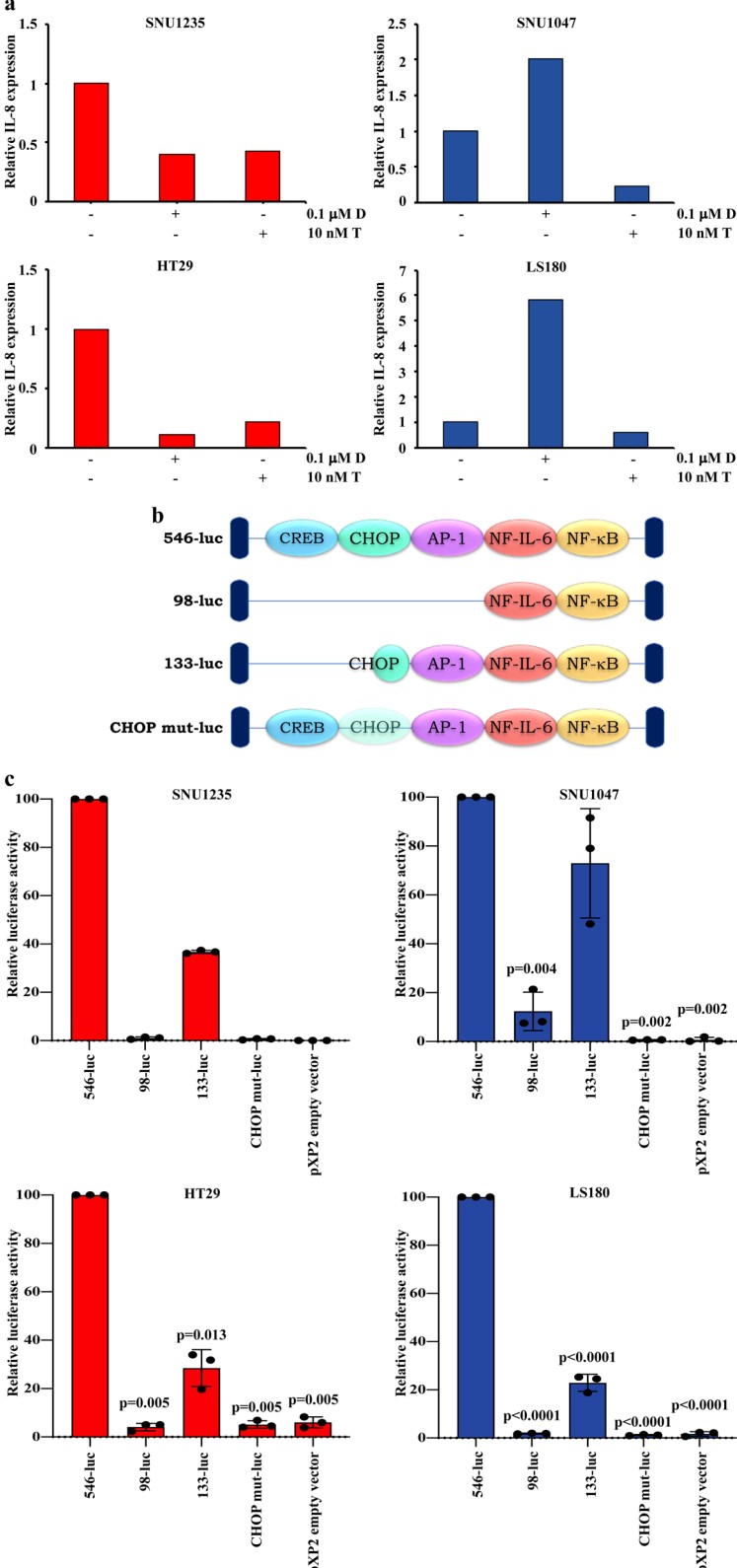

**Fig. 4 CHOP-mediated transcriptional IL-8 regulation. a** SNU1235, SNU1047, HT29, and LS180 cells lines were treated with the indicated concentration of drugs for 2 h and the presence of IL-8 was detected by RT-qPCR in all cell lines. Results were evaluated as ΔΔct of IL-8 relative to RPL19 and expressed as the ratio assuming the levels in the control as 1.0. Results of a representative experiment out of three independent experiments performed are shown. **b** Transcription factor binding regions in the IL-8 gene promoter retained in the different luciferase reporter plasmids. **c** Cell lines were co-transfected with 100 ng luc-reporter vector and 10 ng pRL-TK. Cells were harvested for luc assay 24 h post-transfection. pXP2 empty vector plasmid was used as a control. Results represent the average of three independent experiments. *p*-values indicate statistically significant differences (*p* < 0.05 by 2-tailed Student's *t* test) for the comparison between each reporter construct and the full promoter sequence (546-luc).

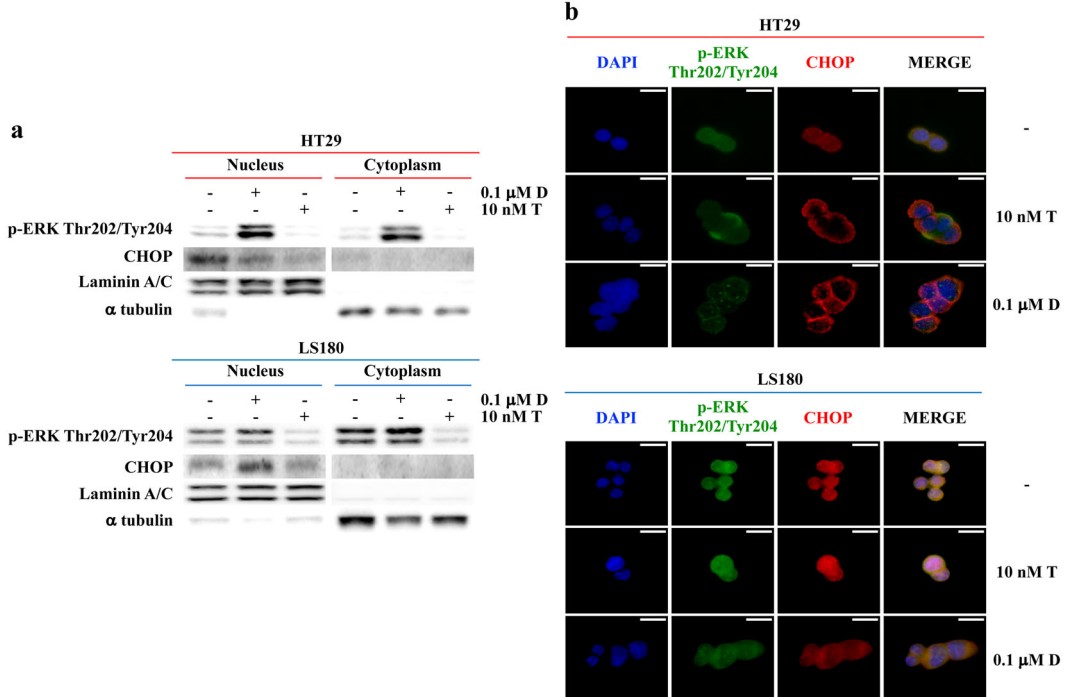

**Fig. 5 CHOP subcellular localization after dabrafenib and trametinib treatment.** HT29 and LS180 cell lines were treated with fixed doses of dabrafenib (D) and trametinib (T) for 24 h, as indicated. **a** Cytoplasmic and nuclear fractions of cell lines were isolated; molecular effects were analyzed by WB using specific antibodies (Laminin A/C and α tubulin are shown as protein loading and blotting control for the nuclear and cytoplasmic compartments, respectively). **b** Direct immunofluorescence analysis of the localization of CHOP (red) and p-ERK Thr202/Tyr204 (green) protein in the cytoplasm and nucleus (blue) of HT29 and LS180 cells, after D and T treatment. Results of a representative experiment out of three independent experiments performed are shown. Scale bars 10 µm.

production in a panel of human CRC cell lines; however, *BRAF* mutations, but not PTEN status, specifically dictated the response of CRC cells to selective pathway inhibitors, particularly MAPK pathway inhibitors, in terms of IL-8 production. Indeed, to the best of our knowledge we describe here for the first time a drug-sensitive, genetic-context-dependent BRAF/ERK2/CHOP molecular axis, tightly controlling IL-8 transcription in CRC models.

IL-8 has recently emerged as a putative prognostic/predictive biomarker in CRC. In preclinical models, IL-8 promotes both tumor and endothelial cell proliferation, migration, angiogenesis and decreases sensitivity to the cytotoxic effects of oxaliplatin[4]. Clinically, IL-8 expression correlates with CRC progression and development of liver metastases and is associated with resistance to antiangiogenic therapy[9,14,15]. Regulation of IL-8 expression occurs at three different levels: repression and activation of the gene promoter, mRNA stabilization, and post-translational cleavage of its precursor[16]. It is now well established that the MAPK pathway is the main regulator of IL-8 expression: ERK and c-Jun N- terminal Kinase (JNK) promote AP-1- and NF-κB-mediated IL-8 transcription, whereas p38 stabilizes IL-8 mRNA[17,18]. Consistent with a dominant role of the MAPK pathway in IL-8 regulation, genetic or pharmacologic MEK inhibition invariably abrogated IL-8 production, regardless of the genetic background of the examined CRC cells[19]. Similarly, ERK2 silencing almost invariably downregulated IL-8 expression. As extensively revised by Buscà, ERK1 and ERK2 are generally described as homologous molecules and seem to be functionally redundant, though differential roles for ERK1 and ERK2 in terms of cell proliferation, colony formation, epithelial-mesenchymal transition and cell invasion have been described[20]. Here we show that only ERK2, but not ERK1, gene silencing results in IL-8 downregulation; similar results have been reported with the gp130 subunit of the

promiscuous IL-6 receptor, whose expression is selectively controlled by ERK2, but not by ERK1[21].

At a difference with MEK/ERK inhibition, pharmacologic BRAF inhibition affected IL-8 differentially, according to *BRAF*-mutational status. Paradoxical downstream MAPK activation has been extensively described in *BRAF*-wt genetic contexts, thus MAPK-dependent downregulation and upregulation of IL-8 production in response to dabrafenib in *BRAF*-mut and *BRAF*-wt contexts is theoretically expected[22–25]. Consistent with the recently reported ability of combined BRAF/MEK inhibition to offset paradoxical MAPK activation in *BRAF*-wt models, the combination of dabrafenib and trametinib effectively prevented dabrafenib-induced IL-8 upregulation in both *BRAF*-wt cell lines SNU1047 and LS180[22]. However, BRAF silencing, alone or in combination with dabrafenib treatment, had strikingly different effects in these two models. In the SNU1047 cell line, BRAF silencing inhibited both constitutive and dabrafenib-induced IL-8 production, consistent with a model in which a kinase-inhibited, but not an absent, BRAF protein can heterodimerize with CRAF and paradoxically activate downstream elements of the MAPK pathway[25]. Conversely, in the LS180 cell line, which harbors a *KRAS*[G12D] mutation, BRAF silencing paradoxically upregulated IL-8 production and combined BRAF silencing and dabrafenib treatment synergistically increased IL-8 levels. In CRC an intricate relationship exists between Epidermal Growth Factor Receptor (EGFR) and RAS family signaling[26]. Although we have not addressed this specific issue experimentally, it is possible that *KRAS*-wt cells (whether *BRAF*-mut or -wt) mostly rely on EGFR signaling to feed both basal and stimulated MAPK activation; in this context, BRAF protein expression and kinase activity may have a more prominent role in the activation of MAPK signaling, as opposed to *KRAS*-mut contexts, where CRAF, but not BRAF, is

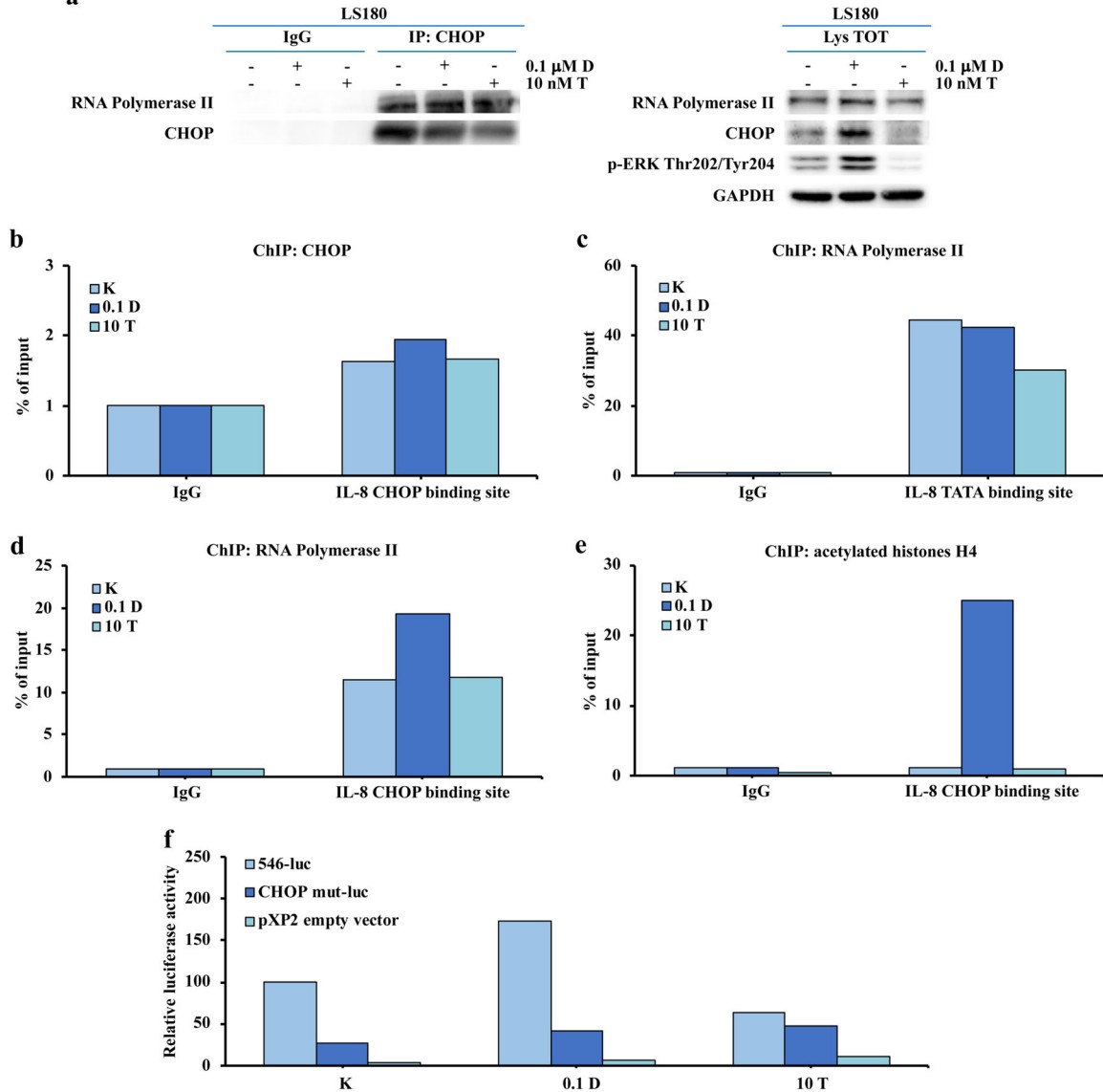

**Fig. 6 CHOP-RNA Polymerase II association to IL-8 gene promoter.** LS180 cell line was treated with fixed doses of dabrafenib (D) and trametinib (T) for 8 h, as indicated. **a** Whole LS180 lysate was immunoprecipitated with either anti-CHOP antibodies or control IgG and analyzed by WB using specific antibodies, as indicated; WB of the whole cell lysate are reported as control. LS180 chromatin was immunoprecipitated with anti-CHOP (**b**), anti-RNA Polymerase II (**c**, **d**) or anti-Acetyl-Histone H4 (**e**) antibodies or with control IgG and analyzed by RT-qPCR with primers specific for CHOP-binding region and TATA box of the IL-8 gene promoter. Results of a representative experiment out of three independent experiments performed are shown. **f** Cell lines were co-transfected with 100 ng luc-reporter vector and 10 ng pRL-TK. Cells were harvested for luc assay 24 h post-transfection and treated with fixed doses of D and T for 24 h, as indicated. pXP2 empty vector plasmid was used as a control. Results of a representative experiment out of three independent experiments performed are shown.

essential to allow the signaling flow to downstream elements of the cascade[22,27]. It is interesting to note that ERK2 silencing is not effective at inhibiting IL-8 production in the *KRAS*-mut LS180 cells and that dabrafenib-stimulated IL-8 production is further increased by ERK1 or ERK2 silencing in *BRAF*-wt contexts. Overall, our data suggest that individual MAPK elements, namely BRAF and ERK2, may play different roles in regulating IL-8 production in *BRAF*-wt CRC cells, depending on *KRAS* and, possibly, EGFR family activation status.

At a difference with previous findings, we observed that NF-κB is important but not sufficient to promote IL-8 transcription in CRC models; however, it has been demonstrated that other transcription factors (such as AP-1) physically interact with NF-κB and functionally cooperate to promote IL-8 gene expression

and might be targeted by MAPK regulation[13,28]. Indeed, our data show an expected modulation of c-Jun Ser73 phosphorylation in response to dabrafenib and trametinib[29]. To the best of our knowledge, here we report for the first time a more prominent role for the CHOP, also known GADD153, transcription factor in MAPK-dependent regulation of IL-8 transcription in CRC models, depending on their *BRAF*-mutational status. In the last decade, several groups demonstrated that CHOP promotes IL-8 gene transcription independently of NF-κB in several cellular contexts, such as T lymphocytes and cystic fibrosis bronchial epithelial cells[30,31]. Here, we report that CHOP resides in the nuclear compartment of untreated CRC cell lines, while treatment with dabrafenib or trametinib modulates CHOP subcellular localization and consequently IL-8 production. In the new model

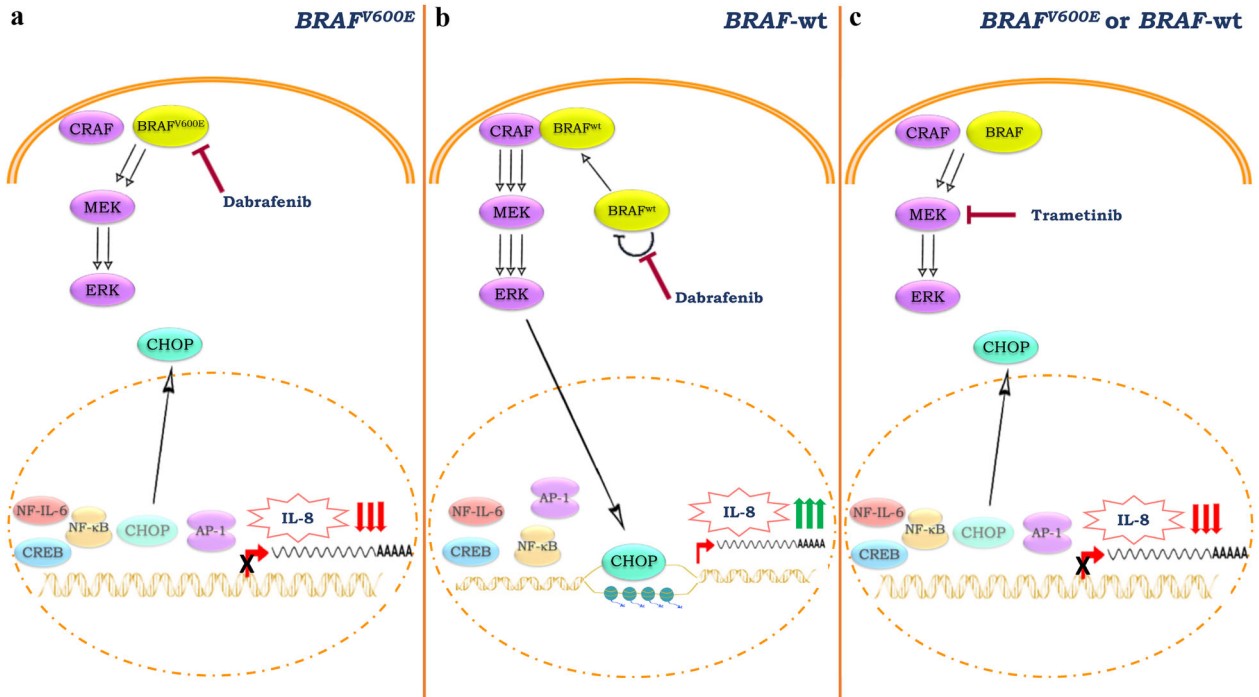

**Fig. 7 Working model of CHOP-dependent IL-8 regulation, after dabrafenib and trametinib treatment, according to BRAF status. a** Upon addition of the BRAF inhibitor dabrafenib, signaling output from *BRAF*$^{V600E}$ is blocked and there is a transient suppression of ERK activation and MAPK signaling. In this genetic context, CHOP is exported to the cytoplasmic compartment, thus resulting in the downregulation of IL-8. **b** In contrast, in *BRAF*-wt contexts, even in the presence of dabrafenib, BRAF forms a complex with CRAF and hyperactivates CRAF itself, thereby driving paradoxical hyper-activation of both MEK and ERK. Due to CHOP nuclear import and histone acetylation, chromatin is accessible to CHOP, thereby increasing its binding to the IL-8 promoter, resulting in IL-8 transcription and protein production. **c** Trametinib inhibits MEK activity, thereby downregulating p-ERK levels: CHOP translocates from the nucleus to the cytosol and IL-8 gene is not transcribed, regardless of *BRAF* genetic status.

(Fig. 7) we propose that CHOP and AP-1 are the key regulators of MAPK-dependent IL-8 gene transcription in CRC: in *BRAF*-wt contexts dabrafenib causes paradoxical ERK activation, nuclear CHOP accumulation and binding to the IL-8 promoter, which, together with increased AP-1 activation, results in increased IL-8 transcription; trametinib, on the other hand, shuts down ERK, AP-1, and CHOP activation, thereby downregulating IL-8 transcription; in *BRAF*-mut cell lines CHOP is retained in the cytoplasm, with a perinuclear distribution, after either BRAF or MEK inhibition, thereby explaining the downregulation of IL-8 expression. Interestingly, Oh and coll. have described a similar situation in which CHOP enhances DR5 transcription after paradoxical MEK/ERK activation induced by BRAF inhibition in *RAS*-mut cell lines[32]. Furthermore, we demonstrated that dabrafenib affects not only CHOP compartmentalization, but also IL-8 gene chromatin accessibility. Indeed, in a *BRAF*-wt context, specific BRAF inhibition increases the binding of the RNA Pol II to the CHOP-binding region in IL-8 promoter, hence resulting in IL-8 transcription. Consistently, the CHOP-binding region displays an increased histone H4 acetylation, which results in an open-chromatin status. As consequence, dabrafenib induces IL-8 promoter activation in a CHOP-dependent manner. Compelling evidence shows that oncogenic signaling controls gene transcription, not only by affecting transcription factors' recruitment, but also by affecting chromatin structure through post-translational histone modifications[33]. Among them, signaling through the classical MAPK module controls histone acetylation and phosphorylation in a kinase-dependent and -independent manner[34]. Interestingly, it has been demonstrated that ERK2 acts as a DNA-binding protein, hence interfering with the binding of transcriptional factors[35].

It is nowadays accepted that soluble factor networks are involved in tumor-stroma interactions; cytokines and chemokines production (including IL-8) may be sustained not only by cancer cells but also by stromal elements (namely fibroblasts, endothelial, and immune cells), in a bidirectional crosstalk[2]. In a complex TME, stromal and/or infiltrating immune cells may substantially contribute to IL-8 expression; IL-8 regulation, in turn, can be modulated by targeted agents, such as selective BRAF inhibitors, differentially in a genetically normal stromal compartment, as compared to the tumor cell compartment, in which the net effect of pathway inhibition appears to be dictated by the specific genetic landscape of the tumor. This should be taken into account in interpreting CRC biology, especially in response to molecularly targeted drugs.

Overall our data depict a complex regulation of IL-8 production in CRC and identify a BRAF/ERK2/CHOP axis which dictates the overall effect of pharmacological MAPK manipulation in different genetic contexts. In addition to response to pharmacological modulation, *BRAF* status deserves further investigation as a potentially predictive biomarkers of IL-8 production *in vivo*, in both tumor and infiltrating mononuclear cells. This, in turn, may help identify the clinical settings in which IL-8 targeting (with IL-8 or CXCR1/2 antagonists currently in clinical development) might be most promising as a therapeutic strategy.

## Methods

**Cell lines**. CRC cell lines were kindly provided from Federica Di Nicolantonio (University of Turin, Turin, Italy)[36]. X-MAN™ HCT116 Parental and HCT116 PTEN$^{−/−}$ were generated by Horizon from homozygous knock-out of PTEN by deleting exon 5 which encodes the active site of the protein in the CRC cell line HCT116 (Horizon Discover, www.horizondiscovery.com)[12]. Isogenic cell lines

HCT116, HK2–6 and HKE-3 and DLD-1, DKO-1 and DKO-4 were performed by Shirasawa's group by gene targeting technique[37].

Cell lines were routinely maintained in RPMI 1640 or DMEM medium supplemented with 10% fetal bovine serum, 2 mM L-glutamine, and antibiotics (Pen/Strep) (all from Euroclone, Milan, Italy) in a humidified atmosphere with 5% $CO_2$ at 37 °C.

All cell lines tested negative for mycoplasma contamination.

**Drug treatments.** Trametinib (GSK1120212) and dabrafenib (GSK2118436) were kindly provided by GlaxoSmithKline (Brentford, Middlesex, UK). Gedatolisib (PF05212384) was kindly provided by Pfizer Inc. (New York, NY, USA). SCH772984 and alpelisib were purchased by Selleckchem (Houston, TX, USA). Everolimus was obtained from Novartis Pharma (Basel, Switzerland). MK-2206 was kindly provided by Merck and Co. (Kenilworth, NJ, USA).

Trametinib, dabrafenib, SCH772984, alpelisib and MK-2206 were dissolved in DMSO as 1 mM, 10 mM, 0.2 mM, 10 mM and 1 mM stock solution, respectively and stored at −20 °C. Gedatolisib was dissolved in DMSO as 1 mM stock solution and stored at −80 °C. Everolimus was dissolved in 100% ethanol as a 10 mM stock solution and stored at −20 °C.

The final concentration of drugs was obtained by dilution with culture medium.

**RNA transfection.** Cells were transfected with either a siRNA against BRAF (sequence: 5′-UACACCAGCAAGCUAGAUGCA-3′), MEK and ERK1 (Santa Cruz Biotechnology, Santa Cruz, CA) or shRNA plasmids against ERK2 (Santa Cruz Biotechnology, Santa Cruz, CA). RNAs were transfected with RNAiMax reagent (Invitrogen, Carlsbad, CA, USA) for 24 h according to the manufacturer's instructions.

**Standardization and assessment of cell culture media.** In order to overcome the possibility that a different number of cells alter the results of chemokines analysis, growth curves for each CRC cell line were assessed and cells were seeded at different cell concentrations to evaluate their rate growth. For cell counting, Thoma chamber was used.

Standardization: all cell lines were plated into 60 × 15 dishes (BD Falcon, Oxford, UK) to have about 1×10⁶ cells after 48 h of plating. After 24 h from the plating, the culture medium was replaced by serum-free medium, and after 24 h, media were collected and cells were counted.

Assessment: CRC cell culture media were analyzed in triplicate as per the manufacturer's instructions using human IL-8 (Enzo Life Sciences, Farmingdale, NY, USA) and VEGF (R&DSystems, MN, USA) specific ELISA. Absorbance was read at 450 nm. IL-8 and VEGF expression was represented as pg/mL and then related to the control. HCT116 Parental and HCT116 PTEN⁻/⁻ culture media were analyzed by Human Angiogenesis Antibody Array (RayBiotech, Norcross, GA, USA), according to the manufacturer's protocol.

**RNA analysis.** Total RNA was prepared from cells using the RNA extraction kit, RNeasy Mini Kit (Qiagen, Hilden, Germany) as per the manufacturer's instructions. Of total RNA, 0.5 or 1 μg was converted into single-strand cDNA using Superscript II (Invitrogen, Carlsbad, CA, USA) as per the manufacturer's instructions. RT-qPCR was performed with Fast SYBR®Green quantitative PCR kit (Applied Biosystems, Foster City, CA, USA) for RPL19 (Forward primer sequence: 5′-CGGAAGGGCAGGCACAT-3′ and Reverse primer sequence 5′-GGCGCAAAA TCCTCATTCTC-3′), IL-8 (Forward primer sequence: 5′-AGGGTTGCCAGATGC AATAC-3′ and Reverse primer sequence 3′-CCTTGGCCTCAATTTTGCTA-5′) and PTEN (Forward primer sequence: 5′-AATAAAGACAAAGCCAACCGATA CTT-3′ and Reverse primer sequence 5′-CGGCTCCTCTACTGTTTTTGTGA-3′). Expression of PTEN and IL-8 mRNA was then normalized with RPL19 and PTEN was compared with mRNA positive control of T98G.

**WB analysis.** Whole cell extracts were obtained by NP-40 lysis buffer, containing 50 mmol/L Tris-HCl (pH 8.0), 250 mmol/L NaCl, 1 mmol/L EDTA, 100 mmol/L NaF, 1 mmol/L NaVO4, 10 mmol/L PMSF, 10 μg/mL leupeptin, 1% NP40. Assay sample for protein concentration used was Bio-Rad Protein Assay Dye Reagent Concentrate (BioRad, Hercules, CA, USA). An amount of total cell lysate was fractionated by SDS-polyacrylamide gel electrophoresis and transferred to nitrocellulose membrane (Amersham, Arlington Heights, USA). Membranes were probed with the following primary antibody: phosphorylated (Thr202/Tyr204) (#4370) and total (#9102) ERK1/2, BRAF (#9433), MEK 1/2 (#9122), CHOP (#2895), PTEN (#9552), phosphorylated NF-κB Ser536 (#3033), phosphorylated CREB Ser133 (#9191), phosphorylated c-Jun Ser63 (#9261) and Ser73 (#9164) (Cell Signaling Technology Inc., Beverly, USA; dil 1:1000); RNA Pol II (#05–623) (clone CTD4H8, Millipore Corporation, Bedford, MA, USA; dil 1:1000). Signal was detected using peroxidase-conjugated anti-mouse or anti-rabbit secondary antibodies (Jackson Immunoresearch Labs, Inc., Baltimore, USA). The enhanced chemi-luminescence system (Amersham, Arlington Heights, USA) was used for detection and image detection was performed with UVITEC Alliance 4.7 system (Cambridge, UK). To control the amount of proteins transferred to nitrocellulose membrane either β-Actin and GAPDH were used and detected by anti-mouse β-Actin antibody (#A1978) (clone AC-15, Sigma-Aldrich, St. Louis, USA; dil 1:200)

and anti-rabbit GAPDH (#5174) (Cell Signaling Technology Inc., Beverly, USA; dil 1:1000), respectively. Laminin A/C (#4777) and α tubulin (#2144) (Cell Signaling Technology Inc., Beverly, USA; dil 1:1000) were used as nuclear and cytoplasmic fractionation control.

Full, uncropped blot/gel images are available in Supplementary Fig. 11.

**CHOP direct-site mutagenesis and luc assay.** CHOP mut-luc vector was constructed by deletion of CHOP-binding site from 546-luc vector with QuikChange II XL Site-Directed Mutagenesis Kit (Agilent Technologies, Stockport, UK). Primers were generated harboring the desired mutation: Forward primer sequence: 5′-CG TATTTGATAAGGAACAAATAGGAAACTCAGGTTTGCCCTG-3′ and Reverse primer sequence 5′-CAGGGCAAACCTGAGTTTCCTATTTGTTCCTTATCAA ATAC-3′. The presence of the correct mutation was verified by plasmid sequencing (Supplementary Fig. 12).

98-luc, 133-luc, 546-luc, NF-κB mut-luc, NF-IL-6 mut-luc, and AP-1 mut-luc vectors were kindly provided from Prof. Naofumi Mukaida (University of Kakuma-machi, Kanazawa, Japan)[38].

The empty vector pXP2 plasmid in Escherichia coli (ATCC) has been extracted with QIAprep Spin Miniprep Kit (Qiagen, Hilden, Germany) following the manufacturer's instructions and used as a control.

SNU1235, SNU1047, HT29, and LS180 cells were transfected with Lipofectamine 3000 following the manufacturer's instructions (Invitrogen, Carlsbad, CA, USA). Cells luc activities were analyzed by the Dual-Glo luc assay system (Promega, Madison, WI, USA) in the GloMax 96 Microplate Luminometer (Promega Madison, WI, USA) after 48 h from transfection.

**Nuclear and cytoplasmic protein extraction.** 4 × 10⁶ cells (SNU1235, SNU1047, HT29, LS180) were resuspended in a hypotonic lysis buffer (10 mM HEPES pH 7.5, 10 mM KCl, 0.1 mM EDTA, 0.1 mM EGTA pH 8, 1 mM DTT) containing protease and phosphatase inhibitors (Thermo Fisher Scientific, Rochester, NY, USA) for nuclear and cytoplasmic extractions. After resuspension, NP-40 was added to a final concentration of 10% and the nuclei were isolated by centrifugation at 10,000 rpm for 30 s at 4 °C. After removing the supernatant (i.e. the cytoplasmic extract), the nuclei were washed for 3 times in hypotonic lysis buffer and centrifugated at 10,000 rpm for 30 s at 4 °C. Nuclei were resuspended in a nuclear extract buffer (20 mM HEPES pH 7.5, 0.4 M NaCl, 1 mM EDTA, 1 mM EGTA, 1 mM DTT, protease and phosphatase inhibitors), rocked for 30 min in ice and then recovered by centrifugation at 14,000 rpm for 10 min at 4 °C. Nuclear and cytoplasmic protein extraction was analyzed by WB.

**Immunofluorescence.** SNU1235, SNU1047, HT29, and LS180 cells were seeded on 22x22 mm coverslips and medium was replaced by serum-free medium after 24 h from plating. After 24 h from the replaced medium, the grown cells were fixed in 4% formaldehyde in Phosphate-Buffered Saline (PBS) for 10 min, permeabilized in PBS containing 0.5% Triton X-100 for 5 min, and blocked with PBS containing 5% Bovine Serum Albumin (BSA) and 0.3% of Triton X-100 for 90 min. Coverslips were incubated overnight with primary antibodies (CHOP and phosphorylated Thr202/Tyr204 ERK1/2, Cell Signaling Technology Inc., Beverly, USA) in 1% BSA, 0.3% Triton X-100 in PBS. Coverslips were washed three times in PBS and then incubated with one of the following secondary antibodies for 1 h at a dilution of 1:400: goat anti-rabbit and goat anti-mouse (A11034 and A11032 respectively, Alexa Fluor®, Invitrogen, Carlsbad, CA, USA). After washing, the nuclei were counterstained with DAPI. Cells were then washed twice with PBS and observed under Zeiss Axiovert 200 M fluorescence microscope at 60X magnification. Specific fields were photographed with a digital camera equipped with Zeiss Axiovision acquisition software.

**IP.** Chip-Grade Protein G Magnetic Beads (Thermo Fisher Scientific, Rochester, NY, USA) were incubate with 5 μg of CHOP (Cell Signaling Technology Inc., Beverly, USA) overnight at 4 °C. Precleared beads were than incubated overnight at 4 °C with 1 μg of protein. The immunoprecipitates were collected and after 2 wash in CHAPS buffer resuspended in 35 μl of the same buffer and Ladder buffer 1X. The immune complexes and 20 μg of protein total cell extract were analysed by WB.

**Formaldehyde cross-linking and ChIP.** Formaldehyde was added directly to cell culture media of 6 × 10⁶ LS180 cells, at a final concentration of 1% at room temperature for 8 min and the cross-linking was stopped by the addition of glycine to a final concentration of 0.125 M. Cells were rinsed twice with cold PBS, incubated with 5 ml of cold PBS containing protease and phosphatase inhibitors (Thermo Fisher Scientific, Rochester, NY, USA), and then scraped. Cells were collected by centrifugation at 1.000 rpm for 10 min at 4 °C. Pellets were resuspended in 5 vol of lysis buffer (5 mM piperazine N, N bis zethone sulfonic acid pH 8, 10 mM KCl, 85 mM KCl, 0.5% NP-040, protease and phosphatase inhibitors) and incubated on ice for 20 min. Nuclei were collected by centrifugation at 3,000 rpm, resuspended in sonication buffer (1% SDS, 10 mM EDTA, 50 mM Tris HCl pH 8) and incubated on ice for 10 min.

Chromatin was sonicated on ice to an average length of 500 bp and then microcentrifuged at 14,000 rpm for 10 min at 4 °C.

Immunoprecipitation was performed with ChIP Grade Protein A/G Magnetic Beads (Thermo Fisher Scientific, Rochester, NY, USA). Magnetic beads were precleared with dilution buffer (0.01% SDS, 1.2 mM EDTA, 16.7 mM Tris HCl pH 8, 1.1% Triton X-100, 167 mM NaCl) and incubated with a mixture containing 4 μg of affinity-purified anti-acetyl-Histone H4 (Lys8) anti-rabbit polyclonal antibody (Cell Signaling Technology Inc. Beverly, USA) or IgG antibody (Santa Cruz Biotechnology, Santa Cruz, CA, USA) overnight at 4 °C with mild shaking. Precleared beads were than incubated overnight at 4 °C with a mixture of chromatin 1:10 with dilution buffer with mild shaking. The immunoprecipitates were collected and after 5 wash in buffer A (0.1% SDS, 2 mM EDTA, 20 mM Tris HCl pH 8, 1% Triton X-100, 150 mM NaCl) and 4 wash in buffer B (0.1% SDS, 2 mM EDTA, 20 Mm Tris HCl pH 8, 1% Triton X-100, 500 Mm NaCl) were eluted with elution buffer (1% SDS, 100 mM NaHCO$_3$) for 30 min at 37 °C with gently rotation.

Input of chromatin was collected before the first wash from the supernatant of each IgG samples and was processed with the eluted immunoprecipitates beginning at the crosslink reversal step.

Samples were centrifuged at 14,000 rpm for 5 min and supernatants were transferred in clean tube. Crosslinks were reversed by addition of NaCl to a final concentration of 200 mM by incubation at 65 °C for 4 h with shaking. 5 μl of Proteinase K Solution (Qiagen, Hilden, Germany) were added to samples and incubated for 1 h at 43 °C.

DNA was extracted with the addiction of 1:1 of phenol:chloroform:isoamyl alcohol (25:24:1), centrifuged at 14,000 rpm for 5 min at room temperature and then precipitated with 1:10 vol of 3 M sodium acetate (pH 5), 10 μg of glycogen and 2.5 vol of 100% ethanol at −20 °C overnight. Pellets were collected by microcentrifugation, resuspended in 30 μl of H$_2$O, and analyzed by using RT-qPCR. Standard curves were obtained from serial dilutions of input control samples (1–1:1000). PCR reactions contained 1 μl of immunoprecipitate or diluted 1:10 total input, primers and Fast SYBR®Green quantitative PCR kit (Applied Biosystems, Foster City, CA, USA) in a total volume of 10 μl. For CHOP PCR analysis, the following oligonucleotides were used: Forward primer sequence: 5′-TCAAAGAAAACTTTCGTCATACTCCG-3′ and Reverse primer sequence 5′-CGATTTGCAACTGATGGCCC-3′.

**DNA extraction**. Genomic DNA was extracted using the QIAamp DNA Mini Kit (Qiagen, Hilden, Germany) according to the manufacturer's instructions. The extracted DNA was quantified and its quality assessed using NanoDrop® (Thermo Fisher Scientific, Rochester, NY, USA) and Qubit® (Thermo Fisher Scientific, Rochester, NY, USA) platforms according to manufacturer' instructions.

**Next generation sequencing (NGS)**. NGS was performed with a panel (the Ion Ampliseq Cancer Hotspot Panel v2, Thermo Fisher Scientific, Rochester, NY, USA) composed of 207 amplicons, covering >2800 hotspot mutations in 50 genes: ABL1, AKT1, ALK, APC, ATM, BRAF, CDH1, CDKN2A, CSF1R, CTNNB1, EGFR, ERBB2, ERBB4, EZH2, FBXW7, FGFR1, FGFR2, FGFR3, FLT3, GNA11, GNAS, GNAQ, HNF1A, HRAS, IDH1, JAK2, JAK3, IDH2, KDR, KIT, KRAS, MET, MLH1, MPL, NOTCH1, NPM1, NRAS, PDGFRA, PIK3CA, PTEN, PTPN11, RB1, RET, SMAD4, SMARCB1, SMO, SRC, STK11, TP53, and VHL.

Multiplex PCR libraries were generated from 10 ng of DNA per sample using the Ampliseq technology (Ion Ampliseq Library Kit v2.0, Thermo Fisher Scientific, Rochester, NY, USA). Each library was barcoded with the Ion Xpress Barcode Adapters 1–16 kit and 17–32 kit (Thermo Fisher Scientific, Rochester, NY, USA). Library concentration was evaluated with Qubit 2.0 fluorometer using high sensitivity Qubit Assay Kit (Thermo Fisher Scientific, Rochester, NY, USA). Each diluted library (100 pM) was clonally amplified on to Ion Sphere Particles (ISP) using emulsion PCR (emPCR) in an Ion Chef System (Thermo Fisher Scientific, Rochester, NY, USA) according to the manufacturer's instructions. Enriched ISPs were loaded on to 530 chips accommodating thirty-two tumor samples on a single chip per sequencing run.

Sequencing was performed on an Ion S5 Sequencer using an Ion 530 Chip and an Ion 530 kit-Chef (all from Thermo Fisher Scientific, Rochester, NY, USA).

**Data analysis and reporting**. The raw data were analyzed using the Torrent Suite software (version 5.10.1) (Thermo Fisher Scientific, Rochester, NY, USA) through default analysis parameters. Variant Caller version 5.10.1.20 and Coverage Analysis version 5.10.0.3 plug-ins (Thermo Fisher Scientific, Rochester, NY, USA) were used for variant calling and sequencing coverage analysis, respectively. A minimum sequencing depth of 250× was considered as adequate sequencing depth, and an allelic frequency of 5% was used as a cut-off for variants.

Ion Reporter™ Server hosting informatic tools (Ion Reporter™ Software version 5.4) was used for variant analysis, filtering, and annotations. The Integrative Genomics Viewer was used to visualize the read alignment and the presence of variants against the reference genome as well as to confirm variant calls by checking for strand biases and sequencing errors.

Only mutations reported in the Sanger Institute Catalogue of Somatic Mutations in Cancer (COSMIC) database (http://www.sanger.ac.uk/cosmic) were considered.

**Statistics and reproducibility**. Results are expressed as average of three independent experiments or as representative experiment out of three independent experiments performed with similar results. Optimal cut-off and performance characteristics (sensitivity, specificity, AUC) were evaluated by computing ROC curves. AUC under 0.70 were not considered as relevant. The associations between variables were tested by two-sided Pearson Chi Square test or Fisher exact test, when appropriate. Mean comparison of more than two groups was made by ANOVA, when appropriate. The SPSS® (21.0), R® (2.6.1), MedCalc® (13.0) statistical programs were used for all analyses. $p$-value of < 0.05 was considered statistically significant.

## Data availability

All data supporting the findings of this study are available in the article along with Supplementary Information files. Source data are available in Supplementary Data 1. Additional information for relevant mutations are reported in Supplementary Table 1.

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

## Acknowledgements

This work was supported in part by grants from "Fondazione AIRC per la Ricerca sul Cancro" (Ludovica Ciuffreda, IG 18622) and "Bando Interno Ricerca Corrente IRE 2018". Fabiana Conciatori and Chiara Bazzichetto were supported by an AIRC fellowship for Italy. The authors wish to thank IRCCS Scientific Director office for supporting the manuscript.

## Author contributions

M.M. and L.C. contributed to conception and design of all the experiments, supervised data acquisition and analysis, and wrote the manuscript. F.C. and C.B. performed experiments, contributed to data acquisition, analysis and interpretation of the results, and wrote the manuscript. C.A.A., S.D., and I.F. performed experiments and contributed to data acquisition and analysis. I.S. performed all statistical analyses. M.G.D., S.B., S.S., G.B., and F.C. provided critical reagents, contributed to conception and critically revised the manuscript. All authors reviewed and gave final approval.

## Competing interests

The authors declare no competing interests.
