## [Peer Review File · Communications Biology]

Reviewers' comments:

Reviewer #1 (Remarks to the Author):

The manuscript entitled 'BRAF status modulates Interleukin-8 expression through a CHOP-dependent mechanism in colorectal cancer' by Conciatori et al, with reference file 3965330, tries to shed light on the molecular mechanisms of IL-8 expression regulation via MAPK and PI3K pathways. They investigated the role of BRAF mutations and PTEN loss in IL-8 production. They observed that IL-8 levels are profoundly affected by the modulation of the MAPK signalling pathway. The manuscript is well written, experiments are appropriate and the conclusions support the results shown.

Minor comments:

- Results, MAPK-dependent regulation of IL-8 expression, page 6, line 149, ...increase after BRAF and ... change to ...increase after BRAF silencing and ...
- Figure 5b; In the main text it is stated that PTEN-loss is correlated with low IL-8 in infiltrating mononuclear cells, which is supported by the figure, but in the legend it is swapped. It should be PTEN-loss /low IL-8 infiltrate and not PTEN loss/high IL-8 infiltrate.

Reviewer #2 (Remarks to the Author):

The authors studied the BRAF/ERK2/CHOP regulatory axis in regulating interleukin-8 production in CRCs. Using a series of cell lines, the authors identified the correlation and function of the molecules, which provide valuable information for clinical application.

However, the pathogenesis of colorectal cancer involves a multi-factorial interaction, and cytokines may play a dual role in CRCs, and tumor cells can be adaptively changed upon the micro-environment. The authors should take caution in conclusion;

As the classical regulation of IL-8 regulation, PI3K/mTOR seems not involved in the present study, is it the specific pattern in certain cell lines or in the pathogenesis of the disease;

Typos should be carefully checked through the context.

Reviewer #3 (Remarks to the Author):

Manuscript Number: COMMSBIO-19-1035-T

The authors showed high levels of IL-8 expression is correlated with BRAF mutation and PTEN loss in colorectal cancer (CRC). Given that IL-8 is a useful biomarker to monitor changes in tumor burden following anticancer therapy, this study has prognostic importance in CRC. The authors provided unequivocal proof that IL-8 expression is controlled by BRAF/MEK/ERK signaling cascade through regulation of CHOP subcellular localization. The experiments are well-designed and well-implemented. Based on the IL-8 immunohistochemistry findings in 168 CRC patients, it is evident that BRAF mutations and PTEN status is not correlated with high level of IL-8 expression in tumor cells. This finding creates confusion and challenges the authenticity of the cell line-based findings by the authors. My concerns about this study are duly noted below:

Major comments:

1. In Figure 2a, authors used two FDA-approved drugs, namely dabrafenib and trametinib to inhibit

BRAFV600E and MEK activity, respectively. Subsequently, authors used another experimental drug by Pfizer, namely gedatolisib as a PI3K/mTOR inhibitor. It is not clear to me why authors did not use FDA-approved PI3K/mTOR inhibitor in this experiment.

2. In Figure 2b and subsequent other bar graphs, authors used two-tailed Student's t-test to measure differences between groups. A t-test is a type of statistic used to determine if there is a significant difference between the means of two groups. Whereas, 1-way ANOVA compares the means of two or more independent groups to determine statistical differences. My strong recommendation to authors is that they should redo the statistical test and report the p-value in all pertinent experiments.

3. In Figure 3, authors described how CHOP protein positively regulates IL-8 expression. They showed how mutation in the DNA binding sites of AP-1 and CHOP regulates IL-8 expression. It has been reported that physical association between AP-1 and NF- κ B promote IL-8 expression (Page 11, Line 10-11). Regulation of IL-8 expression by CHOP is a novel finding. The main focus of this figure is CHOP-mediated regulation of IL-8 transcription. It is confusing when authors showed five different proteins bind to the IL-8 promoter and regulate its expression. I strongly encourage not to include NF- κ B and AP-1 findings in the figure 3 because this data do not include additional information to our existing knowledge.

4. In Figure 4c, authors performed chromatin immunoprecipitation (ChIP) experiment to determine the occupancy of CHOP protein in the IL-8 promoter. The experiment is confusing when authors mentioned that chromatin was immunoprecipitated with anti-Histone H4K8 antibody and analyzed by RT-qPCR (Page 29, Line 7). Authors should perform ChIP experiment using anti-CHOP and anti-RNA Pol II antibodies. The enrichment of the chromatin by CHOP proteins should be reported as percentage of Input. To better design the experiment authors may follow this study (Cell Death Differ. 2016 Apr;23(4):707-22).

5. In Figure 5, authors performed IHC to measure IL-8 expression in 168 human CRC tissue specimens. The outcome of this experiment is confusing because there is no correlation between high IL-8 expression and BRAF/PTEN status. This result challenges the notion that IL-8 can be used as a prognostic biomarker to measure tumor progression following targeted therapies against BRAF or MEK.

6. While explaining the IHC data, authors found that CRC patients harboring PTEN deletion show higher number of penetrated mononuclear immune cells expressing low level of IL-8 compared to PTEN WT patients. It is very confusing to me why PI3K/mTOR/PTEN pathway has no effect on IL-8 transcription in tumor cells however mononuclear immune cells show an opposite impression. Authors should discuss, if necessary, perform experiments to explain the mechanism of IL-8 regulation by PI3K/mTOR/PTEN pathway in mononuclear immune cells.

Minor comments:

1. The axis label of each figure is very small and illegible.
2. Authors should expand the abbreviation in first use.
3. The title of figure 3 needs to be revised.
4. In page 7, line no. 23; authors should change the word from complete to full-length.
5. The information about the statistical test performed in figure 1c is missing.
6. Figure 3b and c are mislabeled.
7. Scale bar is missing in figures 4b and 5b.

REDACTED REBUTTAL FOR TRANSPARENT PEER REVIEW

RE: Manuscript Number: COMMSBIO-19-1035-T

Dear Editor,

enclosed please find a thoroughly revised version of the manuscript entitled “**BRAF status modulates Interleukin-8 expression through a CHOP-dependent mechanism in colorectal cancer**” by Fabiana Conciatori and co-authors.

First, we would like to thank reviewers for their helpful and constructive comments; we carefully took all of them into account, performed additional experiments where necessary, and believe that the revised manuscript has now substantially improved. We thus hope it can be considered for publication in Communications Biology.

A point-by-point reply to individual reviewers' comments follows:

Reviewer #1 (Remarks to the Author):

The manuscript entitled ‘BRAF status modulates Interleukin-8 expression through a CHOP-dependent mechanism in colorectal cancer’ by Conciatori et al, with reference file 3965330, tries to shed light on the molecular mechanisms of IL-8 expression regulation via MAPK and PI3K pathways. They investigated the role of BRAF mutations and PTEN loss in IL-8 production. They observed that IL-8 levels are profoundly affected by the modulation of the MAPK signalling pathway.

The manuscript is well written, experiments are appropriate and the conclusions support the results shown.

We thank the reviewer for his recognition of the potential relevance of our report.

Minor comments:

- Results, MAPK-dependent regulation of IL-8 expression, page 6, line 149, ... increase after BRAF and ... change to ... increase after BRAF silencing and ...

We thank the reviewer for spotting such error, we have now corrected the sentence.

- Figure 5b; In the main text it is stated that PTEN-loss is correlated with low IL-8 in infiltrating mononuclear cells, which is supported by the figure, but in the legend it is swapped. It should be PTEN-loss /low IL-8 infiltrate and not PTEN loss/high IL-8 infiltrate.

We thank the reviewer for spotting such error; however, according to other reviewers' comments, we have now removed *in vivo* results in clinical CRC series and the relative figure from the manuscript (see also response to reviewer #3 below).

Reviewer #2 (Remarks to the Author):

The authors studied the BRAF/ERK2/CHOP regulatory axis in regulating interleukin-8 production in CRCs. Using a series cell lines, the authors identified the correlation and function of the molecules, which provide valuable information for clinical application.

We thank the reviewer for his recognition of the potential relevance of our report.

However, the pathogenesis of colorectal cancer involves a multi-factorial interaction, and cytokines may play dual role in CRCs, and tumor cells can be adaptively changed upon the micro-environment. The authors should take caution in conclusion;

We completely agree with the reviewer that experimental models often oversimplify biological phenomena and that especially modulation of soluble factors' levels might represent adaptive changes driven by tumor/microenvironment interactions. As briefly described in the Introduction section, our interest in IL-8 regulation stems from several lines of evidence highlighting a potential role of this chemokine as a prognostic/predictive factor in CRC; for this very reason our group is currently conducting

a meta-analysis of published data on the prognostic/predictive role of IL-8 levels in published clinical series, whose results will be reported in a separate manuscript. We have, however, given a more balanced and conservative interpretation of the relevance of our results *in vivo* in the complex scenario of clinical CRC, acknowledging potential limitations.

As the classical regulation of IL-8 regulation, PI3K/mTOR seems not involved in the present study, is it the specific pattern in certain cells lines or in the pathogenesis of the disease;

Following reviewer's suggestion, we have now analyzed IL-8 production in response to several different inhibitors of individual elements of the PI3K, AKT, mTOR cascade: modulation of IL-8 expression appears to be variable in response to different inhibitors and not related to the specific genetic background of the cellular model analyzed.

Typos should be carefully checked through the context.

We have thoroughly checked the manuscript for typos and corrected them in the revised version being submitted.

Reviewer #3 (Remarks to the Author):

The authors showed high levels of IL-8 expression is correlated with BRAF mutation and PTEN loss in colorectal cancer (CRC). Given that IL-8 is a useful biomarker to monitor changes in tumor burden following anticancer therapy, this study has prognostic importance in CRC. The authors provided unequivocal proof that IL-8 expression is controlled by BRAF/MEK/ERK signaling cascade through regulation of CHOP subcellular localization. The experiments are well-designed and well-implemented.

We thank the reviewer for his recognition of the potential relevance of our report.

My concerns about this study are duly noted below:

Major comments:

1. In Figure 2a, authors used two FDA-approved drugs, namely dabrafenib and trametinib to inhibit BRAFV600E and MEK activity, respectively. Subsequently, authors used another experimental drug by Pfizer, namely gedatolisib as a PI3K/mTOR inhibitor. It is not clear to me why authors did not use FDA-approved PI3K/mTOR inhibitor in this experiment.

To date, no double PI3K/mTOR inhibitors have been approved by the FDA or EMA. Gedatolisib (PF05212384) was tested in CRC in clinical trials and was in phase I development in a combination study at our Institution at the time this project was started. However, following the reviewer's suggestion, we have now analyzed IL-8 production in response to several different inhibitors of individual elements of the PI3K, AKT, mTOR cascade: modulation of IL-8 expression appears to be variable in response to different inhibitors and not related to the specific genetic background of the cellular model analyzed (see also response to reviewer #2).

2. In Figure 2b and subsequent other bar graphs, authors used two-tailed Student's t-test to measure differences between groups. A t-test is a type of statistic used to determine if there is a significant difference between the means of two groups. Whereas, 1-way ANOVA compares the means of two or more independent groups to determine statistical differences. My strong recommendation to authors is that they should redo the statistical test and report the p-value in all pertinent experiments.

Thank you for the excellent suggestion. We re-analyzed data using the 1-way ANOVA test and modified Figure 2a and Supplementary Fig. 3b of the revised manuscript accordingly.

3. In Figure 3, authors described how CHOP protein positively regulates IL-8 expression. They showed how mutation in the DNA binding sites of AP-1 and CHOP regulates IL-8 expression. It has been reported that physical association between AP-1 and NF- κ B promote IL-8 expression (Page 11, Line 10-11). Regulation of IL-8 expression by CHOP is a novel finding. The main focus of this figure is CHOP-mediated regulation of IL-8 transcription. It is confusing when authors showed five different proteins bind to the IL-8 promoter and regulate its expression. I strongly encourage not to include NF- κ B and AP-1 findings in the figure 3 because this data do not include additional information to our existing knowledge.

Thank you for the useful suggestion. As suggested by the Reviewer, the role of NF- κ B and AP-1 in IL-8 expression is well recognized, thus we moved these data in the Supplementary Fig. 4 of the revised manuscript, as an internal control of our experimental system. Figure 4 of the main manuscript now only focuses on CHOP role in IL-8 production.

4. In Figure 4c, authors performed chromatin immunoprecipitation (ChIP) experiment to determine the occupancy of CHOP protein in the IL-8 promoter. The experiment is confusing when authors mentioned that chromatin was immunoprecipitated with anti-Histone H4K8 antibody and analyzed by RT-qPCR (Page 29, Line 7). Authors should perform ChIP experiment using anti-CHOP and anti-RNA Pol II antibodies. The enrichment of the chromatin by CHOP proteins should be reported as percentage of Input. To better design the experiment authors may follow this study (Cell Death Differ. 2016 Apr;23(4):707-22).

Additional ChIP experiments using anti-CHOP and anti-RNA Pol II antibodies, as well as additional IP experiments showing physical association between CHOP and activated RNA Pol II under different experimental conditions are now being completed and will be included in the revised manuscript.

[REDACTED]

[REDACTED]

[REDACTED]

[REDACTED]

Minor comments:

1. The axis label of each figure is very small and illegible.

We appreciate this suggestion and modified all Figures accordingly.

2. Authors should expand the abbreviation in first use.

All abbreviations are now expanded at their first use.

3. The title of figure 3 needs to be revised.

We corrected the title of the revised Figure 4 into “*CHOP-mediated transcriptional IL-8 regulation*”.

4. In page 7, line no. 23; authors should change the word from complete to full-length.

We corrected the revised manuscript as suggested.

5. The information about the statistical test performed in figure 1c is missing.

Thanks for spotting this error, information on the statistical test used is now included in the revised Figure legend.

6. Figure 3b and c are mislabeled.

Revised Figure 4 (replacing previous Figure 3) has been modified as suggested.

7. Scale bar is missing in figures 4b and 5b.

Thank you for spotting this error, we added the scale bar in immunofluorescence images (revised Figures 5b and Supplementary Figure 5b).

REVIEWERS' COMMENTS:

Reviewer #2 (Remarks to the Author):

The authors have addressed my concerns very well.

Reviewer #3 (Remarks to the Author):

I have no additional comments.